# Hybrid spin Hall nano-oscillators based on ferromagnetic metal/ferrimagnetic insulator heterostructures

Haowen Ren [1] ✉, Xin Yu Zheng [2], Sanyum Channa[3], Guanzhong Wu[1], Daisy A. O'Mahoney[4], Yuri Suzuki [2] & Andrew D. Kent [1] ✉

Spin-Hall nano-oscillators (SHNOs) are promising spintronic devices to realize current controlled GHz frequency signals in nanoscale devices for neuromorphic computing and creating Ising systems. However, traditional SHNOs devices based on transition metals have high auto-oscillation threshold currents as well as low quality factors and output powers. Here we demonstrate a new type of hybrid SHNO based on a permalloy (Py) ferromagnetic-metal nanowire and low-damping ferrimagnetic insulator, in the form of epitaxial lithium aluminum ferrite (LAFO) thin films. The superior characteristics of such SHNOs are associated with the excitation of larger spin-precession angles and volumes. We further find that the presence of the ferrimagnetic insulator enhances the auto-oscillation amplitude of spin-wave edge modes, consistent with our micromagnetic modeling. This hybrid SHNO expands spintronic applications, including providing new means of coupling multiple SHNOs for neuromorphic computing and advancing magnonics.

High-efficiency oscillators are essential to accelerate the application of spintronics for neuromorphic computing[1–4], Ising systems[5], and magnonic devices[6–8], among other applications. Spin-Hall nano-oscillators (SHNOs) are one of the important approaches to achieve these applications due to their two-dimensional geometry, which permits coupling multiple SHNOs in a plane[9–11], as well as their ease of fabrication. Several geometries of SHNOs have been proposed in previous studies, such as nanodisk[12–14], nanowire[15–17], and nanoconstriction types[18–22]. However, these SHNOs generally have high threshold currents, low emission powers, and poor quality factors because of the nature of their constituent materials, specifically the large magnetic damping in transition metal ferromagnets. In recent years, attention has focused on ferrimagnetic insulators[23–26] due to their extremely low damping and, consequently, high magnon conductivity[27], which is very favorable for spintronic applications. At the same time, this low damping characteristic facilitates the formation of spin-Hall effect-induced auto-oscillations; indeed, ferrimagnetic insulator-based nano-

oscillators have been demonstrated with yttrium iron garnet/Pt bilayers[25,26]. Nevertheless, they suffer from low power emission due to their small inverse spin-Hall effect signals[28]. Joule heating also limits their application at room temperature[28]. One way to overcome these drawbacks is by creating a new type of hybrid SHNOs based on ferromagnetic metal-ferrimagnetic insulator heterostructures. Interesting physics emerges when coupling thin layers of these two types of materials[29–32]. When the two layers are weakly coupled, there are two distinct spin resonances, associated with acoustic and optical modes. However, when they are strongly coupled, the two layers act collectively, leading to magnetic properties inherited from both layers[33], specifically a lower effective damping. Thus, SHNOs fabricated from such heterostructures can take advantage of the low damping from the ferrimagnetic insulator layer and yet maintain a strong electrical signal from the ferromagnetic metal layer.

Theoretical studies have shown that a uniform spin current applied to an extended magnetic thin film does not support the

[1]Center for Quantum Phenomena, Department of Physics, New York University, New York, NY 10003, USA. [2]Department of Applied Physics and Geballe Laboratory for Advanced Materials, Stanford University, Stanford, CA 94305, USA. [3]Department of Physics and Geballe Laboratory for Advanced Materials, Stanford University, Stanford, CA 94305, USA. [4]Department of Materials Science and Engineering and Geballe Laboratory for Advanced Materials, Stanford University, Stanford, CA 94305, USA. ✉e-mail: haowren@gmail.com; andy.kent@nyu.edu

formation of auto-oscillations due to the emergence of nonlinear damping from magnon-magnon interactions[34]. However, by concentrating spin current in a small region, linear spin-wave mode auto-oscillation states can be stabilized[35]. Later, it was shown that nonlinear localized modes[36] could also be achieved due to the suppression of magnon-magnon interactions, which has been experimentally demonstrated in point-contact type and disk-type SHNOs[12–14,37]. Meanwhile, if the device geometry is confined (e.g., a nanowire or nanoconstriction), auto-oscillations can still be excited in a localized region that leads to a potential well that limits spin-wave propagation[16,20]. These self-localized modes can have much smaller threshold currents than linear modes due to lower radiative loss in an auto-oscillation state.

In this article, we demonstrate a new type of SHNO that combines ferromagnetic transition metals $Ni_{80}Fe_{20}$ (Py) with an epitaxial thin film ferrimagnetic insulator, lithium aluminum ferrite (LAFO). This hybrid SHNO expands spintronic applications, including providing new means of coupling multiple SHNOs for neuromorphic computing and can advance designs for magnonics. Furthermore, compared to conventional Py/Pt SHNOs, this hybrid SHNO is superior in all important characteristics, having a reduced threshold current, stronger emission power, and higher quality factor.

## Results and discussion

Our heterostructures are composed of two different lithium aluminum ferrite compositions ($Li_{0.5}Al_{1.0}Fe_{1.5}O_4$ (LAFO) or $Li_{0.5}Al_{0.5}Fe_2O_4$ (LFO)) ($x$ nm)/Py(5 nm)/Pt(5 nm) layers with varied LAFO or LFO thickness $x$ (including $x = 0$, i.e., just Py/Pt layers). The Py/Pt layers are patterned into 400 nm wide nanowires with a 400 nm gap between two Au contact pads, as shown in Fig. 1a. Detailed deposition and fabrication conditions are in Methods. We fabricated devices with LAFO: LAFO4/Py5/Pt5, LAFO10/Py5/Pt5, and LAFO20/Py5/Pt5, and LFO: LFO15/Py5/Pt5 with the numbers being the layer thicknesses in nm. Lastly, a Py5/Pt5 reference device was deposited on a sapphire substrate.

To determine the magnetic properties in different Py/LAFO samples, ferromagnetic resonance spectroscopy (FMR) measurements were carried out on unpatterned thin films and heterostructures via a vector network analyzer (VNA) technique[38]. Effective magnetization $M_{eff}$ and anisotropy field $H_a$ are obtained by fitting resonance peaks to the Kittel model $f = \mu_0 \gamma / 2\pi \sqrt{(H + H_a)(H + H_a + M_{eff})}$, where $H$ is the

external magnetic field, $M_{eff}$ is the effective magnetization, $\gamma$ is the gyromagnetic ratio, and $\mu_0$ is the vacuum permeability. Both LAFO and LFO have magnetocrystalline anisotropy with an easy axis along <110> and a hard axis along <100> directions that is characterized by an in-plane anisotropy field $H_a$. Gilbert damping constants $\alpha$ are obtained by measuring and fitting the FMR linewidth as a function of the frequency. The data and fits are shown in Fig. 1b, c and the fitting parameters are listed in Supplementary Table S1. There was always only one FMR absorption peak observable, indicating that the two magnetic layers in our Py/LAFO heterostructures are strongly coupled. Further, the $M_{eff}$ of Pt/Py/LAFO falls between the $M_{eff}$ of bare LAFO or LFO and Py layers, as expected for two ferromagnetically coupled magnetic layers. To analyze the change of $M_{eff}$ and $\alpha$ in the heterostructures, a macrospin model based on Landau–Lifshitz–Gilbert (LLG) equation considering two strongly coupled magnetic layers is used (see Supplementary Note 7). When two magnetic layers are strongly ferromagnetic coupled, the acoustic mode resonance condition will be set by the weighted mean of the magnetic properties of the two individual layers, $\bar{M}_{eff} = (t_{Py}M_{s,Py}M_{eff,Py} + t_{LAFO}M_{s,LAFO}M_{eff,LAFO})/(t_{Py}M_{s,Py} + t_{LAFO}M_{s,LAFO})$ and $\bar{\alpha} = (t_{Py}M_{s,Py}\alpha_{Py} + t_{LAFO}M_{s,LAFO}\alpha_{LAFO})/(t_{Py}M_{s,Py} + t_{LAFO}M_{s,LAFO})$, where $\bar{M}_{eff}$ and $\bar{\alpha}$ are the weighted effective magnetization and damping constant of the bilayers, $t_{Py}(t_{LAFO})$ and $M_{s,Py}(M_{s,LAFO})$ are the thickness and saturation magnetization of the Py(LAFO) layer, respectively, consistent with previous models of coupled layers[33]. The measured values are listed in Supplementary Table S1 and are compared to the model's $\bar{M}_{eff}$ and $\bar{\alpha}$. $\bar{M}_{eff}$ obtained from the simple model is always smaller than the actual measured value, while $\bar{\alpha}$ is always larger than the measured value, which indicates that the exchange coupling is smaller than that of the weighted mean of the two layers. As the damping of the coupled magnetic layers decreases, a spin current injected into the Py layer in LAFO/Py/Pt heterostructures can excite a larger magnetic volume, which, as we show, greatly improves device performance.

To compare the magnetic excitations of thin films with patterned structures, we conducted spin-torque FMR (ST-FMR) on both 2 μm wide stripe devices and 400 nm wide nanowire devices. Figure 2a, b show the ST-FMR spectra of 400 nm wide nanowire devices. In contrast to the FMR spectra, ST-FMR shows two dominant resonances, whose linewidth and peak amplitude are sensitive to bias current. While for 2 μm width stripe devices, only one ST-FMR peak is seen (Supplementary Fig. S2a).

Figure 2c shows the ST-FMR frequency-resonance field spectra of a 400 nm nanowire and a 2 μm stripe LAFO20/Py5/Pt5 device. We find that the dispersion of the higher frequency mode of the two devices overlap, with a fit to the Kittel model giving $\mu_0 M_{eff} = 0.86$ T, close to that found from the FMR spectra of the associated unpatterned film. We thus attribute this feature to a bulk mode (BM), a spin excitation that is most uniform across the width of the device. The lower frequency mode only appears in the 400 nm wide device and is associated with a much lower $\mu_0 M_{eff} = 0.65$ T. We attribute this lower frequency mode to an edge mode (EM), as indicated in Fig. 2a, b, and this conclusion is supported by micromagnetic simulations, as discussed below. Two modes of this type have been reported in previous studies[15–17,20].

Hybrid SHNO devices show the onset of auto-oscillations at a threshold current. Figure 2d–g shows the power spectral density (PSD) as a function of bias current at fixed field $H = 0.0817$ T for $\phi = 70°$. In all the devices, the auto-oscillation frequency redshifts with increasing bias current, a characteristic of localized modes in nanowire SHNOs[16,17]. This is not a pure heating effect (see Supplementary Fig. S4). Interestingly, the threshold current $I_{th}$ drops dramatically between the reference sample, Py5/Pt5, and the sample with LAFO, LAFO4/Py5/Pt5, and then the $I_{th}$ slowly increases with the thickness of LAFO. The slow increase of $I_{th}$ with the thickness of the LAFO layer agrees well with the prediction of the macrospin model

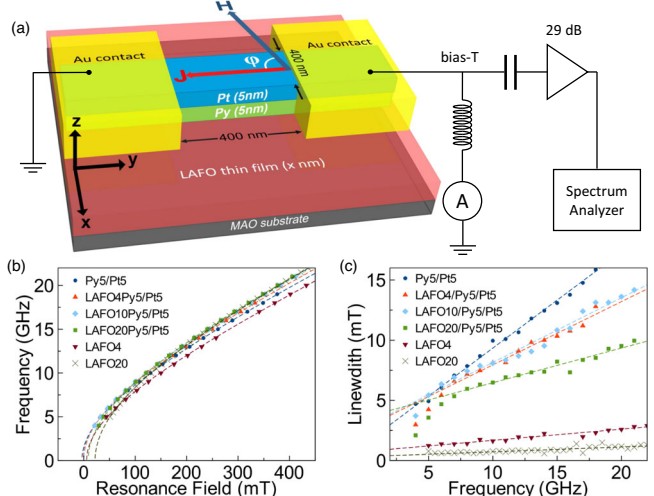

**Fig. 1 | Hybrid nano-oscillator and FMR characteristics. a** Schematic of the hybrid SHNO device and power spectral density (PSD) measurement setup. **b** FMR frequency versus resonance field for various unpatterned thin films and heterostructures, including, for reference, Py5/Pt5 bilayers and LAFO and LFO thin films. **c** FMR linewidth as a function of frequency for these samples.

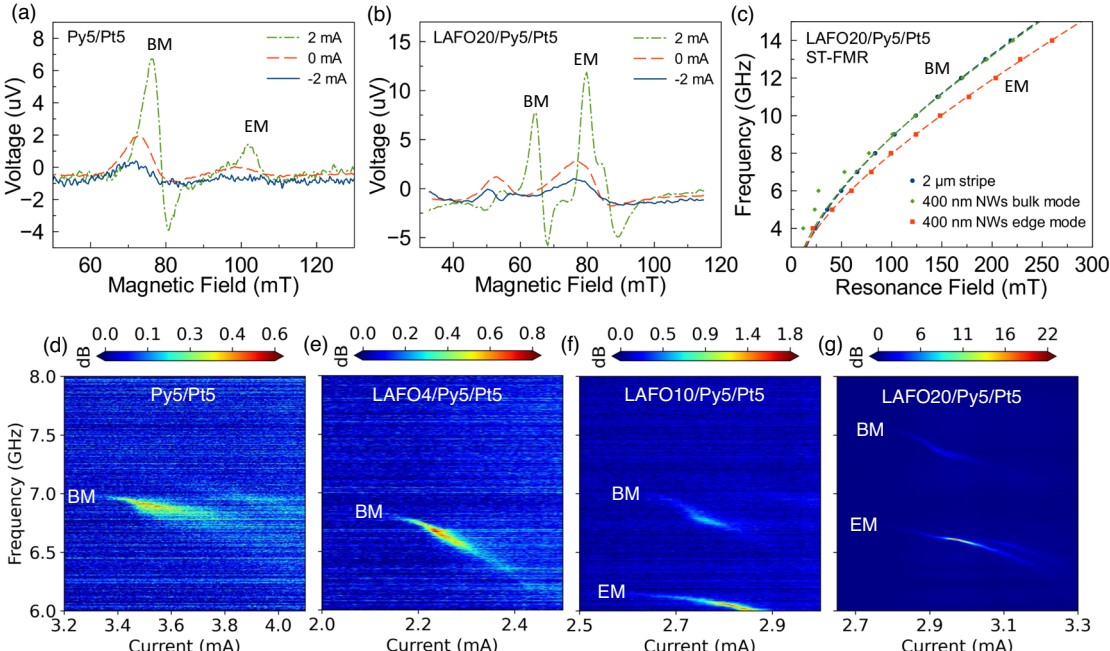

**Fig. 2 | ST-FMR and PSD measurements. a** ST-FMR measurements of 400 nm nanowire device for **a** Py5/Pt5 and **b** LAFO20/Py5/Pt5 at 7 GHz with the applied field at an angle $\phi$=70° with respect to the long axis of the wire. **c** Kittel model fitting curves of LAFO20/Py5/Pt5 for 2 μm stripe (blue circles) device and 400 nm nano-wire device. The green diamond shows fit for the peaks from bulk mode and red square for the peaks from edge mode. Maps of PSDs as a function of frequency and dc bias at a fixed magnetic field $H = 0.0817$ T and $\phi = 70°$ for nanowire devices consisting of **d** Py5/Pt5, **e** LAFO4/Py5/Pt5, **f** LAFO10/Py5/Pt5, and **g** LAFO20/Py5/Pt5. The output power increases significantly for the thickest LAFO sample studied, as indicated by the color scales above each PSD map.

$I_{th} \propto (\alpha_{\mathrm{Py}} t_{\mathrm{Py}} M_{s,\mathrm{Py}} + \alpha_{\mathrm{LAFO}} t_{\mathrm{LAFO}} M_{s,\mathrm{LAFO}}) \bar{M}_{\mathrm{eff}}$, and is consistent with the ST-FMR results obtained from the 2 μm width stripe and 400 nm width nanowire samples (Supplementary Fig. S2b, c). However, the drop of $I_{th}$ from Py5/Pt5 to LAFO4/Py5/Pt5 cannot be explained by this model. We note this decrease in $I_{th}$ is observed in ST-FMR studies conducted on both 400 nm nanowire and 2 μm width strip samples, showing that it does not depend sensitively on sample geometry. It is thus possible that the spin current generated from the Py layer itself acts on the LAFO to increase the spin torques and reduce $I_{th}$[39–42]; previous studies have experimentally shown that spin-orbit torque can be generated from a single magnetic layer[19,43]. In addition, the edge mode can be dominant in hybrid devices and cause a mode-related change of nonlinear damping[44,45], which would reduce radiative loss and thus reduce $I_{th}$. Nevertheless, further study is required to explain this drop of $I_{th}$.

Notice that compared to the Py5/Pt5 sample, the slopes of the redshift increase in all LAFO samples. This is likely due to larger spin-precession angles and the emergence of a nonlinear self-localized mode[18]. Interestingly, the relative magnitude of EM and BM measured from ST-FMRs and PSDs both follow the same trend: the dominant mode transitions from a BM in Py5/Pt5 to an EM in LAFO20/Py5/Pt5, which simultaneously increases the performance of the oscillators. This transition will be discussed in detail in the next section. Threshold currents and auto-oscillation currents extrapolated from the ST-FMR data and determined directly from PSD for different devices are listed in Supplementary Table S2.

To investigate the spin-wave modes of Py and Py/LAFO SHNO heterostructures, micromagnetic simulations were carried out using MuMax3 (see Methods)[46]. Spin currents were applied solely to the 400 nm Py nanowire's central region to mimic the device's current distribution. The simulation is run until a steady state response is observed. The time evolution of magnetization was then converted to the frequency domain by Fast Fourier transform (FFT). Figure 3a shows the spatial-averaged FFT amplitude in the center region of Py for dif-ferent samples, confirming the experimental observation of two dominant auto-oscillation modes. From these simulations, we can find that the auto-oscillation frequencies and their trends are in excellent agreement with our experimental results: (i) the resonance frequency is almost not changed from the Py5/Pt5 device to the LAFO4/Py5/Pt5 device, then increases with the thickness of LAFO. This is mainly due to variations in the net $\bar{M}_{\mathrm{eff}}$, as the resonance frequency strongly depends on $\bar{M}_{\mathrm{eff}}$; this shift of resonance frequency closely follows the trends in $\bar{M}_{\mathrm{eff}}$ determined by FMR and ST-FMR measurements, as shown in Supplementary Table S1; (ii) the high-frequency mode has a higher amplitude in Py5/Pt5, while the low-frequency mode gradually becomes dominant with increasing LAFO thickness; (iii) the peak amplitude and quality factor for LAFO20/Py5/Pt5 is significantly higher than that of Py5/Pt5. To identify the reason behind this transition, pixel-wise spatial FFTs were conducted on Py5/Pt5 and LAFO20/Py5/Pt5 simulations. Spatial FFT images of the Py layer from Py5/Pt5 at the auto-oscillation frequencies are shown in Fig. 3b. We find that the low-frequency peak at $f = 6.21$ GHz is concentrated on the edges of the Py stripe, while the peak at $f = 7.17$ GHz is dominant in the middle. This observation is consistent with our assignment of the modes in the previous section and earlier studies[15,16].

More interesting magnetic behavior occurs in the simulation of hybrid SHNO devices. Spatial FFT images of Py and LAFO layers from the LAFO20/Py5/Pt5 simulation are shown in Fig. 3c. Compared to Py5/Pt5, the EM and BM of the Py layer excited in LAFO20/Py5/Pt5 shows a much larger oscillation amplitude. The magnetization of the LAFO layer oscillates coherently with that of the Py layer. The increased auto-oscillation amplitude is mainly due to a larger precession cone angle in the Py layer, caused by the lower effective damping constant. Mean-while, the area of the EM expands in the bilayer system, which leads to a larger excited volume of moments, consistent with a higher quality factor. The out-of-plane expansion of both BM and EM is caused by the strong ferromagnetic coupling between the two layers, while the in-plane expansion of EM can be understood in this way: the exchange field generated by the LAFO layer tends to align the moments at the edge of Py nanowires against the demagnetization field, which

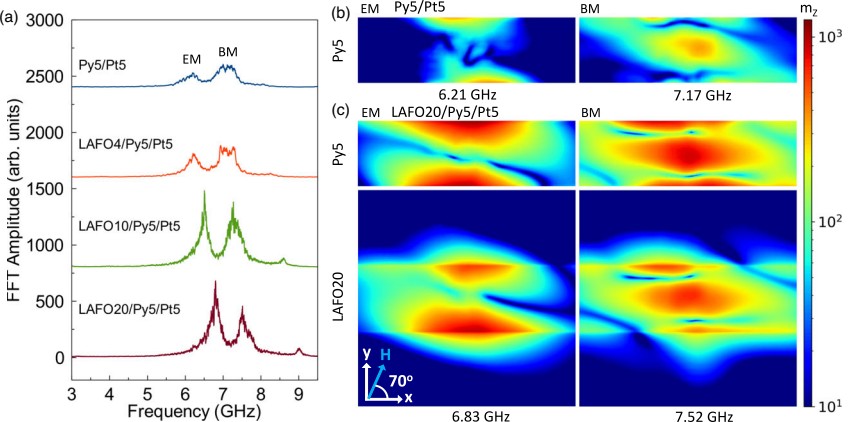

**Fig. 3 | Micromagnetic modeling. a** FFT amplitude spectrum as a function of frequency from micromagnetic simulations of different devices. The spectrum is acquired by doing FFT on the time revolution of spatial-averaged magnetization $\bar{m}_z(t)$ in the center region of the nanowire excited by a spin current. **b** Top views of spatial FFT images on the Py layer of Py5/Pt5 obtained at EM and BM resonance frequencies. **c** Top view of spatial FFT images of the Py layer (top) and LAFO layer (bottom) of a LAFO20/Py5/Pt5 device. The image size for the Py5 layer is 1500 × 400 nm² and for the LAFO20 layer is 1500 × 1500 nm². The logarithmic color scale is on the right, where the color represents the FFT amplitude of $m_z(x,y)$.

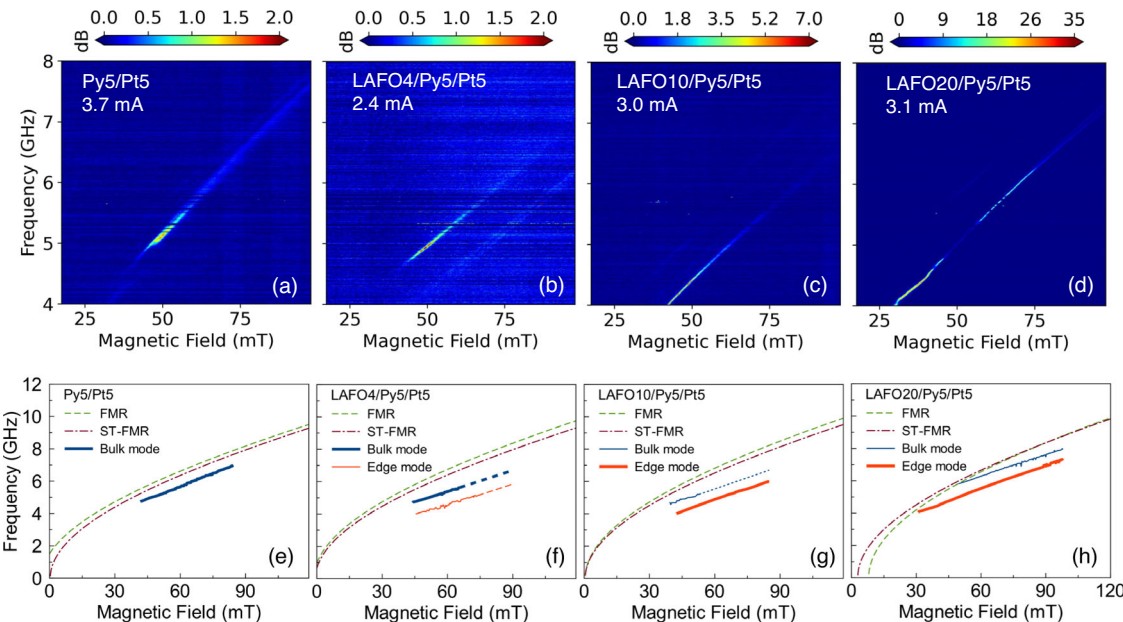

**Fig. 4 | PSD maps as a function of magnetic field and frequency.** PSD maps for **a** Py5/Pt5, **b** LAFO4/Py5/Pt5, **c** LAFO10/Py5/Pt5, and **d** LAFO20/Py5/Pt5 SHNOs. Resonance frequency as a function of external magnetic field for the device **e** Py5/Pt5, **f** LAFO4/Py5/Pt5, **g** LAFO10/Py5/Pt5, and **h** LAFO20/Py5/Pt5 obtained by FMR (brown dash-dot), ST-FMR (green dash), and PSD (blue and red solid) measurements. Dominant resonance modes are highlighted by a wider line. Notice the resonance frequency obtained from PSD has two distinctive peaks, which are associated with bulk mode and edge modes.

decreases the effective field inhomogeneity, thus increasing the EM coherent oscillation volume. This is confirmed by plotting the transverse magnetization profile of the devices at equilibrium (Supplementary Fig. S3a), where one can observe a smoother transition near the edge of Py layer in the LAFO20/Py5/Pt5 device. This expands the area of the localized mode, especially the EM, greatly enhancing the coherence of each mode and thus increasing the maximum power and quality factor of signals emitted from the oscillators.

To better understand the properties of the auto-oscillation modes, maps of the PSD as a function of the magnetic field at a fixed bias current (1.15 times $I_{th}$) are plotted in Fig. 4a–d. Similar to the current dependent PSD map (Fig. 2d–g), BM and EM are observed. By fitting the two PSD peaks to Lorentzian functions, we can obtain the auto-oscillation frequencies and dispersion curves for the BM and EM

as shown in Fig. 4e–h. This is shown in comparison to the FMR and ST-FMR results. In the Py5/Pt5 sample (Fig. 4a), only the BM is detectable, and its dispersion curve is slightly redshifted compared to that of FMR and ST-FMR data. However, contrary to the commonly seen self-localized mode[13,36] in magnetic thin films, in a magnetic wire, the propagation of spin waves is restricted in the transverse direction but allowed along the wire direction, preventing mode localization. This BM has a similar $I_{th}$ compared to the uniform mode[35], which is confirmed by the ST-FMR on 2 μm stripes (Supplementary Fig. S2b). Instead, the EM is localized due to a self-induced potential well, leading to a localized quasi-linear auto-oscillation mode.

In LAFO/Py/Pt hybrid samples (Fig. 4b–d), due to the strong coupling between two magnetic layers, the center region of LAFO will precess coherently with Py. The exchange fields generated from the

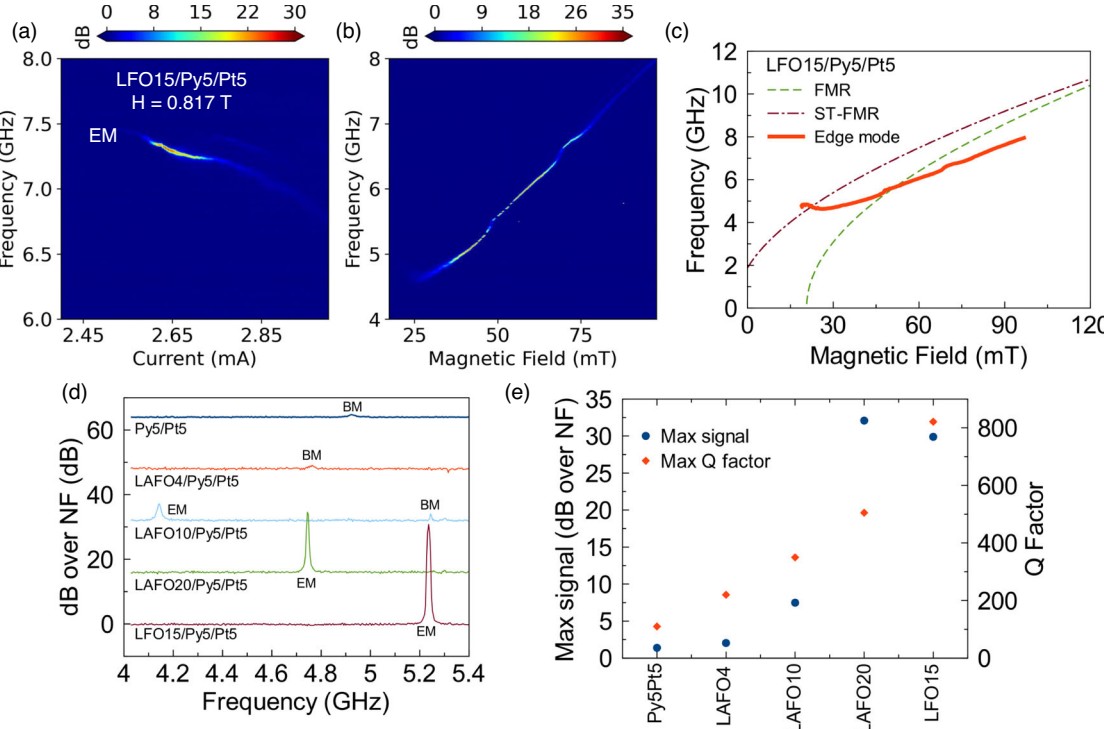

**Fig. 5 | PSD of LFO compared to LAFO and Py SHNOs.** PSD maps for LFO15/Py5/
Pt5 as a function of **a** bias current and **b** magnetic field. **c** Dispersion curves of
LFO15/Py5/Pt5 obtained from FMR, ST-FMR, and PSDs measurements. **d** PSD
spectrums of different samples at fixed $H = 0.045$ T. Lines are shifted upward 15 dB
for each spectrum **e** Max signal (dB over NF) and max Q factor for different devices
obtained from the PSDs in (**a**) and Fig. 4a–d.

LAFO layer change the auto-oscillations frequencies in LAFO/Py/Pt
samples. The increased difference between the dispersion curves of
auto-oscillation and those obtained from FMR and ST-FMR supports
the idea that the spin-wave modes are more localized in the LAFO-
containing devices. As shown in Fig. 4e–h, compared to the Py5/
Pt5 sample, the dispersion curve of the BM of the LAFO4/Py5/
Pt5 sample from PSD measurements is much lower than the FMR
mode. This is one of the key characteristics of a mode that is more
strongly localized.

According to the discussion in the previous sections, we deter-
mined a few critical properties of the ferrimagnetic insulator layer
which can guide us to design a better hybrid SHNOs: (i) low $\alpha$ ferri-
magnetic insulator to have lower $\bar{\alpha}$, (ii) higher $M_{eff}$ to make the auto-
oscillation EM more localized and (iii) appropriate thickness to not
increase the threshold current. To meet these criteria, LFO (15 nm),
which contains a higher Fe concentration compared to LAFO, pos-
sesses these critical properties (as listed in Supplementary
Table S1) and was used in place of LAFO as the ferrimagnetic insulator
in our designed device. As shown in Fig. 5a, b, strong auto-oscillation
signals up to 30 dB over the noise floor (NF) are detected, associated
with EMs. Compared to the LAFO samples, the auto-oscillations occur
at a higher frequency due to the larger $M_{eff}$ of the LFO layer. Figure 3c
is the dispersion curves obtained from FMR, ST-FMR, and PSD mea-
surements for LFO15/Py5/Pt5 sample. Since FMR spectra are measured
at $\phi = 0°$ (magnetic hard axis) and the ST-FMR spectra are measured at
$\phi = 70°$ (closer to the magnetic easy axis direction), a significant dif-
ference between these results occurs due to the large in-plane crys-
talline anisotropy of the LFO layer. The strong anisotropy also causes a
crossing between the maximum PSD signal and ST-FMR curves at low
fields in Fig. 5c. At this low field region, the sample is not magnetically
saturated as we are not measuring the PSD with the field along the easy
axis of LFO. This leads to multidomain states at low field and a reso-
nance frequency that does not dependent monotonically on the
applied field. However, at a higher field, the PSD dispersion curve is

closer to what we obtained from ST-FMR. The auto-oscillation dis-
persion in this field range is redshifted relative to ST-FMR dispersion,
which again indicates the formation of localized auto-oscillation
modes. To systematically compare the performance of different sam-
ples, PSDs at a fixed field $H = 0.045$ T obtained from the field-
dependent PSD maps are plotted in Fig. 5d. Clearly, with the optimi-
zation of the ferrimagnetic insulator layer, the maximum power
in hybrid samples can be at least a 1000 times larger than that of the
Py5/Pt5 sample. We note that the anisotropic magnetoresistance
(AMR) does not vary significantly in the different samples (see Sup-
plementary Fig. S1) and is thus not an important factor in the change of
device output power. By fitting the dominant peak of the PSDs with a
Lorentzian function, we obtained both the maximum power and
maximum quality factor from each device, as shown in Fig. 5e. From all
these results, compared to the conventional Py/Pt SHNOs, we can
obtain orders of magnitude higher emission power and quality factor
in the hybrid SHNOs. In addition, the feasibility of generating
and transporting spin waves in the low damping insulator layer pro-
vides a new platform for applications of SHNOs.

In summary, our work presents a new hybrid type of SHNOs,
which shows superior performance compared to conventional Py/Pt
spin oscillators. In hybrid SHNOs, much higher power emission and
quality factors can be obtained relative to conventional Py/Pt SHNOs.
To understand the mechanism behind the improved performance,
ferromagnetic resonance measurements, and micromagnetic simula-
tions were carried out on both conventional Py/Pt SHNO and hybrid
SHNOs. Results show that the two magnetic layers precess coherently
in bulk and edge modes. Meanwhile, the localization of auto-
oscillations reduces the threshold current and makes the edge mode
the dominant power emission source rather than the bulk mode.
Further, by designing the composition and thickness of the ferrimag-
netic insulator layer, we successfully fabricated hybrid SHNOs with
better performance by replacing LAFO with LFO. Our work expands
the possibility of SHNOs for many types of spintronic applications,

such as synchronizing electrically isolated SHNOs, for neuromorphic computing, Ising machines and magnonic logic circuits.

## Methods

**Sample deposition and fabrication.** Epitaxial $Li_{0.5}Al_{1.0}Fe_{1.5}O_4$ and $Li_{0.5}Al_{0.5}Fe_2O_4$ films are grown on (001) $MgAl_2O_4$ (MAO) substrates at 400 °C in 15 mTorr $O_2$ at a laser fluence of 1.9 J/cm$^2$ by pulsed laser deposition. The deposition of epitaxial $La_{0.5}Al_{1.0-x}Fe_{1.5+x}O_4$ with different compositions $x$ follows the previous study[47,48]. After the growth of the ferrite thin films of various thicknesses and compositions, Py(5 nm)/Pt(5 nm) bilayers are deposited on LAFO and LFO via a Kurt Lester magnetron sputtering system at room temperature. The reference sample Py5/Pt5 is deposited on a c-sapphire (0001) substrate. The as-deposited samples are then spin-coated with PMMA 495 4A and exposed by an Elionix 50 keV E-beam lithography system for the nanowires' patterning. After the development process, the samples are transferred to the Kurt Lester system for Ar plasma dry etching. Residual resists were removed in a ultrasonic cleaner before waveguides with 400 nm gaps between two contact pads are created by E-beam lithography. Finally, Cr(5 nm)/Au(50 nm) contacts are deposited.

Experimental techniques. VNA-FMR is used for detecting the thin film samples' ferromagnetic resonance. For pure LAFO samples, a field-modulated technique is used to detect the low linewidth resonance peaks. The samples are always mounted with the dc magnetic field applied along the in-plane [100] hard axis ($\varphi = 0°$) of the LAFO thin film. ST-FMR measurements are carried out in a probe station with the external field always applied at $\varphi = 70°$ with respect to the current direction. The field is modulated with a coil and the signal is detected by a lock-in amplifier. The DC bias is applied via a Keithley 2400. PSDs are measured via Keysight N9030B spectrum analyzer with a noise floor extension option. Input signals are amplified by an internal 29 dB low-noise amplifier. During the measurement, the resolution bandwidth is always kept at 1 MHz. The noise floor in this setup is −125 dBm. To exclude sample-to-sample variations of resistances, ST-FMRs, and PSD maps, additional samples with the same geometry and composition are measured and shown in Supplementary Note 6.

Micromagnetic simulations. Micromagnetic simulations are run by the Mumax3 micromagnetic simulator[46]. The mesh size is set to $300 \times 300 \times 5$, and each cell size is $5 \times 5 \times 5$ nm$^3$. This length is smaller than the exchange length of Py and LAFO. The top Py layer is designed as a stripe in the center with dimension $400 \times 1500 \times 5$ nm$^3$, while the bottom LAFO layer is extended to the boundaries and varied in thickness. Periodic boundary conditions along the long axis of the Py nanowire are used to eliminate the demagnetization field from the end of the Py stripe. The exchange constant between the Py and LAFO layers is taken to be half of the harmonic mean of the two layers. To reduce the spin-wave reflection at the boundary, we set an exponentially increased damping region near the boundaries of the simulated region. The applied spin current is restricted to the center region of the Py nanowire with dimension $500 \times 400 \times 5$ nm$^3$, since most of the spin current is concentrated between the two Au contacts. Threshold currents are found by running simulations over 200 ns and slowly increasing the applied current until $m_z$ starts to converge to a stable auto-oscillation state. In order to determine the auto-oscillation spectrum in the frequency domain, we set the current to be 1.2 times the threshold current found above and run the simulation for 500 ns. We use FFT algorithms to convert the magnetization evolution in the time-domain to the frequency domain. This method can be used to generate the auto-oscillation spectrum of the full device using the spatial-averaged $\bar{m}_z(t)$ or to generate the spatial profile of each auto-oscillation modes from $m_z(\boldsymbol{r}, t)$. Simulation details for different samples are summarized in Supplementary Note 4, and used parameters are listed in Supplementary Table S3.

## Data availability

The datasets generated during and/or analyzed during the current study are available in Supplementary Materials and also available from the corresponding authors on reasonable request.

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

## Acknowledgements

This research was supported by the Quantum Materials for Energy Efficient Neuromorphic Computing (Q-MEEN-C), an Energy Frontier Research Center funded by the US Department of Energy (DOE), Office of Science, Basic Energy Sciences (BES), under Award DE-SC0019273. Work at Stanford is supported by the US Department of Energy, Director, Office of Science, Office of Basic Energy Sciences, Division of Materials Sciences and Engineering under Contract No. DESC0008505 (X.Y.Z.). S.C. was supported by the Air Force Office of Scientific Research under Grant No. FA 9550-20-1-0293. D.A.O. was supported by the National Science Foundation under award DMR-2037652.

## Author contributions

H.R., Y.S., and A.D.K. conceived the experiment, X.Y.Z., S.C., and D.A.O synthesized the LAFO and LFO thin films and performed part of the FMR characterization, while H.R. deposited the Py and Pt thin films. H.R. fabricated the SHNOs, performed the transport experiments and analyzed the data, including FMR, ST-FMR, and PSD data. G.W. and H.R. performed the micromagnetic simulations. The manuscript was prepared by H.R. and A.D.K. in consultation with all other authors. All authors read and commented on the manuscript.

## Competing interests

The authors declare no competing interests.
