## [Peer Review File · Nature Communications]

Reviewers' Comments:

Reviewer #1:

None

Reviewer #2:

Remarks to the Author:

This manuscript reports the hybrid SHNOs based on ferromagnetic coupling FMI and metal Py as the FM layer. They find that these SHNOs have a higher power emission and better quality factor. By measuring microwave generation spectra and running micromagnetic simulations on a series of samples, the author believes the dominant mode is the localized edge mode rather than the previously observed bulk mode in the pure nanowire Py/Pt-based SHNOs. The results reported in this manuscript are new and of some interest to the magnetism and spintronics community. In brief, this paper provides valuable information to design a better SHNO by tailoring the FM hybrid materials for the application of spintronic devices. But the following several serious issues and my concerns about the experiments and interpretation of the results need to be completely solved.

1. The FMR and ST-FMR of these devices and films (LAFO4/Py5/Pt5, LAFO10/Py5/Pt5, LAFO20/Py5/Pt5, and Py5/Pt5) show that the frequency vs. field dispersion relations exhibit a good consistency for all four samples (e.g., see Fig.4). However, in Fig.5(c), the f vs. H dispersions obtained by FMR and ST-FMR for LFO15/Py5/Pt5 shows a significant deviation. What is the reason?
2. Microwave generation spectra in Fig.2 and Micromagnetic simulations in Fig.3 show that the frequency of the bulk mode increases with inserting LAFO and increasing its thickness. However, one can see that this trend is reversed in Fig.5(d). Are these contradictions come from the different devices?
3. In SM, the authors used the linewidth of ST-FMR as a function of DC bias current to determine the threshold current for LAFO4/Py5/Pt5, LAFO10/Py5/Pt5, LAFO20/Py5/Pt5, LFO15Py5/Pt5, and Py5/Pt5. From Fig.S2(b), one can see that the LAFO4/Py/Pt has the minimum threshold current and the minimum resistance. And other hybrid bilayers with comparable resistance have a higher threshold current than the pure single Py, which is consistent with the deviation (Eq.16) in Supplementary Note 1. However, the critical current I_{\max} corresponding to the max auto-oscillation power (Table S2) exhibits an entirely different trend from that obtained in ST-FMR. Contrary to the analysis of damping in Supplementary Note 1 and ST-FMR, the Py/Pt has the maximum critical current I_{\max} corresponding to its maximum auto-oscillation power. What is the real reason for this inconsistency (or anomaly)? I am concerned that this inconsistent trend may come from the deviation of different devices. It would be better for the authors to estimate the uncertainty by checking different series devices with the same FM layer.

4. In the Method section, the authors should provide all parameters used in their micromagnetic simulations, e.g., M_s , H_{eff} , damping constant, the bulk and interfacial exchange stiffness for Py, LAFO, and LFO, and the spin Hall angle for Pt. Also, the value of the driven current for the obtained results in Fig.3 should be provided.

5. In Fig. 5c, the frequency of the edge mode becomes above the frequency (BM or EM) obtained by ST-FMR for the sample LFO15/Py/Pt5 when the magnetic field is low. What is the reason? In addition, in the LFO15/Py/Pt5 sample, does the bulk mode in microwave generation spectra still observe?

6. There are typos, e.g., in line 175, where "Figs. 3e-h" should be "Figs. 4e-h".

Reviewer #3:

Remarks to the Author:

This manuscript reports that a new hybrid type of spin Hall nano-oscillators (SHNOs, LAFO or LAO/Py/Pt) shows much higher power emission and quality factor in comparison to conventional Py/Pt SHNOs. The results are interesting and the underlying mechanism of such improved performance is well explained based on systematic measurements and micromagnetic simulations. I find this is an interesting result as SHNOs may find use in neuromorphic computing and advancing magnonics.

I have a suggestion. One of the important findings of this work is the drop of threshold current I_{th} from Py/Pt to LAFO/Py/Pt. I believe it is important to explain the drop of I_{th} in this manuscript because a lower I_{th} is also of crucial importance for application and also for understanding the underlying physics. I assume that micromagnetic simulation cannot capture this drop. It means that something is missed in the model. How about considering a spin Hall current generated within Py [see PRB 99, 220405(R)] and a resulting spin-orbit torque on LAFO? Although I'm not sure if this additional spin Hall current from Py can explain the drop of I_{th} , it must be present in this experiment and thus is valuable to check.

I would support publication of this manuscript in NCOMM once the authors address the drop of I_{th} .

REVIEWER COMMENTS

Reviewer #2 (Remarks to the Author):

This manuscript reports the hybrid SHNOs based on ferromagnetic coupling FMI and metal Py as the FM layer. They find that these SHNOs have a higher power emission and better quality factor. By measuring microwave generation spectra and running micromagnetic simulations on a series of samples, the author believes the dominant mode is the localized edge mode rather than the previously observed bulk mode in the pure nanowire Py/Pt-based SHNOs. The results reported in this manuscript are new and of some interest to the magnetism and spintronics community. In brief, this paper provides valuable information to design a better SHNO by tailoring the FM hybrid materials for the application of spintronic devices. But the following several serious issues and my concerns about the experiments and interpretation of the results need to be completely solved.

We appreciate the reviewer's positive comments on the impact of our paper on spintronics and improving spin Hall nano-oscillators.

1. The FMR and ST-FMR of these devices and films (LAFO4/Py5/Pt5, LAFO10/Py5/Pt5, LAFO20/Py5/Pt5, and Py5/Pt5) show that the frequency vs. field dispersion relations exhibit a good consistency for all four samples (e.g., see Fig.4). However, in Fig.5(c), the f vs. H dispersions obtained by FMR and ST-FMR for LFO15/Py5/Pt5 shows a significant deviation. What is the reason?

The difference between the FMR and ST-FMR dispersions for LFO15/Py5/Pt5 is associated with the large in-plane magnetic anisotropy of LFO (compared to LAFO). First, we note --- as we discuss in the Methods section (page 10) --- that in our FMR experiments the magnetic field is applied along the [100] direction ($\varphi=0^\circ$). However, in our ST-FMR experiments, the applied magnetic field is at angle of $\varphi=70^\circ$ (needed to have a large ST-FMR response associated with the magnetoresistive signal, AMR). This is the same field angle ($\varphi=70^\circ$) that we use to measure the PSD maps (i.e., in Figs. 2(d-g), 3(a-d) & 5(a,b)).

LAFO and LFO have an easy axis along $\langle 110 \rangle$ directions and a hard axis along $\langle 100 \rangle$ directions. Thus, in our FMR experiments, the field is applied along the [100] direction or a hard axis direction. While in the ST-FMR setup, the field is applied closer to an easy axis direction. This leads to a difference between the FMR and ST-FMR dispersions. In the case of LFO the anisotropy field H_a is ~ 50 mT and the difference is quite apparent (Fig. 5(c)). While for LAFO H_a is less than about 20 mT (see Table S1 for FMR determination of H_a for these samples) and the difference between the FMR and ST-FMR dispersion curves is much smaller (Fig. 4). We have added text on page 8 to explain this deviation between the results obtained from these two techniques:

2. Microwave generation spectra in Fig.2 and Micromagnetic simulations in Fig.3 show that the frequency of the bulk mode increases with inserting LAFO and increasing its thickness. However, one can see that this trend is reversed in Fig.5(d). Are these contradictions come from the different devices?

We thank the reviewer for pointing this out. First, after rechecking the figure, we need to apologize: In Fig. 5(d), the label for the peak at 4.1GHz from the LAFO10/Py5/Pt5 device should be edge mode, EM (not bulk mode, BM). The BM for the LAFO10/Py5/Pt5 sample overlaps with the high EM peak from the LFO15/Py5/Pt5 device. We have modified Fig. 5(d) to correct the labeling and to make the LAFO10/Py5/Pt5 BM peak more visible. It should thus be clear that the spectral trends in the LAFO(x)/Py/Pt are generally in accord with the simulation results shown in Fig. 2.

More specifically, from the Kittel equation for the FMR dispersion, we know that the resonance frequency depends sensitively on M_{eff} . As shown in Table S1, the M_{eff} of the Py5/Pt5 thin film is 0.78 T, and the M_{eff} of the pure LAFO(4/10/20nm) samples are 0.71 T, 0.94 T, and 1.03 T, respectively. The net M_{eff} of ferromagnetically coupled Py/LAFO bilayer falls in between these two values. Since LAFO4 has a smaller M_{eff} than Py5/Pt5, we expect the resonance frequency to decrease slightly in going from Py5/Pt5 to LAFO4/Py5/Pt5 sample. However, the M_{eff} of thicker LAFO (10 and 20 nm thick) is larger than that of Py5/Pt5 and the trend, as the referee notes and as we find in experiment, is that the frequency of the bulk mode then increases with LAFO thickness, in accord with our simulation results.

We have corrected the label in Fig. 5(d) and added comments in the revised manuscript to address the reason for this general trend of resonance frequency from different samples in the Micromagnetic modeling section at the bottom of page 5 and top of page 6.

3. In SM, the authors used the linewidth of ST-FMR as a function of DC bias current to determine the threshold current for LAFO4/Py5/Pt5, LAFO10/Py5/Pt5, LAFO20/Py5/Pt5, LFO15Py5/Pt5, and Py5/Pt5. From Fig.S2(b), one can see that the LAFO4/Py/Pt has the minimum threshold current and the minimum resistance. And other hybrid bilayers with comparable resistance have a higher threshold current than the pure single Py, which is consistent with the deviation (Eq.16) in Supplementary Note 1. However, the critical current I_{max} corresponding to the max auto-oscillation power (Table S2) exhibits an entirely different trend from that obtained in ST-FMR. Contrary to the analysis of damping in Supplementary Note 1 and ST-FMR, the Py/Pt has the maximum critical current I_{max} corresponding to its maximum auto-oscillation power. What is the real reason for this inconsistency (or anomaly)? I am concerned that this inconsistent trend may come from the deviation of different devices. It would be better for the authors to estimate the uncertainty by checking different series devices with the same FM layer.

The reviewer points to an interesting trend in the ST-FMR determined threshold current, namely a maximum I_{th} in Py/Pt samples, and in LAFO/Py/Pt an I_{th} that increases with increasing LAFO thickness. Further, there is a difference between current at which there is a maximum in the output power I_{max} and I_{th} determined by ST-FMR measurements.

We start by addressing the difference between I_{max} and I_{th} . There are several reasons: 1) First, we note these measured quantities probe different responses. The ST-FMR linewidth vs. current extrapolated to zero linewidth provides the threshold current I_{th} associated with the linear instability of a bulk spin-wave (BM) mode. 2) Auto-oscillations are a nonlinear effect. 3) In addition, in the 400nm width samples with LAFO underlayers, edge modes (EM) provide the largest PSD signal. As we note in the main text, these self-localized modes can have much smaller threshold currents than linear modes due to lower radiative loss in an auto-oscillation state. And, we indeed observe that in LAFO/Py/Pt 400 nm nanowires the I_{max} can be less I_{th} determined by ST-FMR measurements, see Table S2.

Now, to the trends in I_{th} . Our model (Eq. 16) predicts that the ST-FMR I_{th} should increase with increasing LAFO thickness in hybrid samples. We observe an increase in the threshold with LAFO thickness, consistent with the proposed model. However, the I_{th} in Py/Pt is larger than the devices with a LAFO underlayer. The precise reason is not clear but it is possible that spin current generated from the Py layer itself acts on the LAFO to increase the spin torques and reduce I_{th} (see our response to Reviewer #3's comments as well).

We now include a more discussion of these points (highlighted in yellow in the main text and Supplementary Material).

We have also conducted ST-FMR measurements on 400 nm nanowires and observe the same trends we found for 2 μ m wide strips. We further show data taken on two different 400 nm samples with the same layer stacks (Fig. R1 below and also in the Supplementary Information, Fig. S2c), showing that the trends are robust and not associated with sample-to-sample variations.

Figure R1. Linewidth as a function of the bias current obtained from ST-FMR measurements on 400 nm NW samples.

To further address the reviewers last point about the possibility of variation in sample properties, we show below a series of resistance, ST-FMR, and PSD measurements of additional devices. First of all, we measured resistances on different devices as shown in Table R1. We noticed that there is variation in the resistance within the devices that have the same geometry, especially the NW samples. This is mainly because of variations in the contact resistance in these two-terminal devices. Thus, as shown in Fig. R1 and Fig. R2, the resistance variations due to this contact resistance has negligible impact on the ST-FMR results from devices of the same geometry and composition. Finally, the PSD maps as a function of bias current for the additional samples are measured as shown in Fig. R3. The I_{max} obtained from the additional samples and their trend is consistent with previous samples. Thus, it is clear that sample-to-sample variation is not affecting our conclusions.

We have modified the manuscript to show the ST-FMR difference between stripe and NW samples and show the overall consistency between I_{th} obtained from the 400nm nanowire samples' ST-FMR measurements and I_{max} obtained from PSDs measurement on page 4. Extrapolated I_{th} from nanowire samples are added to Table S2. We also included the data from the additional devices in the Supplementary Materials to exclude sample-to-sample variations as important to the trends in I_{max} and I_{th} in our experiments.

Table R1 below shows a summary of the sample variance measurements

Sample	Resistance NWs (Ω)	Resistance stripes (Ω)
Py5/Pt5	124&160	194&203
LAFO4/Py5/Pt5	116&206	163&178
LAFO15/Py5/Pt5	194&220	191&196

Figure R2. Left: ST-FMR dispersion curves. Right: Linewidth as a function of bias current curves. Both are obtained from the 2 μm stripe samples

Figure R3. PSD maps as a function of bias current for the additional samples. The I_{max} obtained from the spectrum analyzer are consistent with the sample with the same composition.

4. In the Method section, the authors should provide all parameters used in their micromagnetic simulations, e.g., M_s , H_{eff} , damping constant, the bulk and interfacial exchange stiffness for Py, LAFO, and LFO, and the spin Hall angle for Pt. Also, the value of the driven current for the obtained results in Fig.3 should be provided.

We have added a Table S3 in the Supplementary Materials, which has all the parameters we used in the simulations for different devices.

5. In Fig. 5c, the frequency of the edge mode becomes above the frequency (BM or EM) obtained by ST-FMR for the sample LFO15/Py/Pt5 when the magnetic field is low. What is the reason? In addition, in the LFO15/Py/Pt5 sample, does the bulk mode in microwave generation spectra still observe?

We thank the reviewer for this comment on the auto-oscillation frequency at low fields. This response is due to the fact that the in-plane magnetic anisotropy in LFO15 is much larger than that of the other devices (i.e., LAFO/Py/Pt and Py/Pt devices; see our response to comment 1). FMR measurements show that the in-plane magnetic anisotropy field in LFO15 and LFO15/Py5/Pt5 is around 53 mT and 22 mT, respectively. As we are not measuring PSD with the field along the easy axis of LFO, the sample is not magnetically saturated in the low field. This leads to multidomain states at low field and a resonance frequency that does not dependent monotonically on applied field.

After carefully checking the raw data from PSD maps, we did not observe the BM in the LFO15/Py5/Pt5 sample. This means that EM is dominant in LFO15/Py5/Pt5 sample and may indicate that the BM in this composition requires more current or larger spin torques to be excited.

6. There are typos, e.g., in line 175, where “Figs. 3e-h” should be “Figs. 4e-h”.

Thank you for noticing this typo; we have corrected this in the revised manuscript. We also thank the reviewer for his/her careful reading of our manuscript and thoughtful comments.

Reviewer #3 (Remarks to the Author):

This manuscript reports that a new hybrid type of spin Hall nano-oscillators (SHNOs, LAFO or LAO/Py/Pt) shows much higher power emission and quality factor in comparison to conventional Py/Pt SHNOs. The results are interesting and the underlying mechanism of such improved performance is well explained based on systematic measurements and micromagnetic simulations. I find this is an interesting result as SHNOs may find use in neuromorphic computing and advancing magnonics.

We are happy with this reviewer’s positive comments on our research and its interest in neuromorphic computing and magnonics.

I have a suggestion. One of the important findings of this work is the drop of threshold current I_{th} from Py/Pt to LAFO/Py/Pt. I believe it is important to explain

the drop of I_{th} in this manuscript because a lower I_{th} is also of crucial importance for application and also for understanding the underlying physics. I assume that micromagnetic simulation cannot capture this drop. It means that something is missed in the model. How about considering a spin Hall current generated within Py [see PRB 99, 220405(R)] and a resulting spin-orbit torque on LAFO? Although I'm not sure if this additional spin Hall current from Py can explain the drop of I_{th} , it must be present in this experiment and thus is valuable to check.

We thank the reviewer for this comment. Indeed, the observation of this drop in I_{th} is interesting and it is an important result of our studies. As the reviewer notes, this is not captured in our micromagnetic modeling. We now point this out in the main manuscript and indicate possible reasons for this discrepancy, including the possible mode transition as we discussed in the response to the Reviewer #2 and that there can be spin currents generated by the Py. We also have included the reference suggested by the referee, PRB 99, 220405(R) (2019) along with additional relevant references.

I would support publication of this manuscript in NCOMM once the authors address the drop of I_{th} .

Reviewers' Comments:

Reviewer #2:

Remarks to the Author:

The authors' response is almost satisfactory. However, the underlying physics about the drop of threshold current I_{th} from Py/Pt (with the dominated bulk mode) to LAFO/Py/Pt (with the dominated edge mode) may be related to two reasons: self-generated spin-orbit torque within Py (which have been experimentally proved, see Nat. Commun. 10, 2362 (2019), PRB 105, 224417 (2022)) or/and the mode-related nonlinear damping (Nat. Commun. 10, 5211(2019), Phys. Rev. Applied 17, 064047(2022)). These closely related references should also be mentioned and cited for discussing the drop of threshold current in the revised manuscript.

Reviewer #3:

Remarks to the Author:

The authors actually did not address the I_{th} drop but instead added some speculations. To my view, it is fine because a solid explanation about the I_{th} drop may not be a main aim of this work and other results are still convincing and useful for the community. I thus support publication as is.

REVIEWER COMMENTS

Reviewer #2 (Remarks to the Author):

The authors' response is almost satisfactory. However, the underlying physics about the drop of threshold current I_{th} from Py/Pt (with the dominated bulk mode) to LAFO/Py/Pt (with the dominated edge mode) may be related to two reasons: self-generated spin-orbit torque within Py (which have been experimentally proved, see Nat. Commun. 10, 2362 (2019), PRB 105, 224417 (2022)) or/and the mode-related nonlinear damping (Nat. Commun. 10, 5211(2019), Phys. Rev. Applied 17, 064047(2022)). These closely related references should also be mentioned and cited for discussing the drop of threshold current in the revised manuscript.

We appreciate the reviewer's positive comments on the response. We now include the references above in the revised manuscript in our discussion of the decrease of threshold current in going from Py/Pt to LAFO/Py/Pt.

Reviewer #3 (Remarks to the Author):

The authors actually did not address the I_{th} drop but instead added some speculations. To my view, it is fine because a solid explanation about the I_{th} drop may not be a main aim of this work and other results are still convincing and useful for the community. I thus support publication as is.

We appreciate the reviewer's comments and recommendation that our manuscript is suitable for publication.